

# Coarse-grained curvature tensor on polygonal surfaces

**Charlie Duclut[1⋆], Aboutaleb Amiri[1†], Joris Paijmans[1] and Frank Jülicher[1,2,3‡]**

**1** Max-Planck-Institut für Physik komplexer Systeme,
Nöthnitzer Str. 38, 01187 Dresden, Germany
**2** Center for Systems Biology Dresden, Pfotenhauerstr. 108, 01307 Dresden, Germany
**3** Cluster of Excellence Physics of Life, TU Dresden, 01062 Dresden, Germany

⋆ duclut@pks.mpg.de, † aamiri@pks.mpg.de, ‡ julicher@pks.mpg.de

## Abstract

Using concepts from integral geometry, we propose a definition for a local coarse-grained curvature tensor that is well-defined on polygonal surfaces. This coarse-grained curvature tensor shows fast convergence to the curvature tensor of smooth surfaces, capturing with accuracy not only the principal curvatures but also the principal directions of curvature. Thanks to the additivity of the integrated curvature tensor, coarse-graining procedures can be implemented to compute it over arbitrary patches of polygons. When computed for a closed surface, the integrated curvature tensor is identical to a rank-2 Minkowski tensor. We also provide an algorithm to extend an existing C++ package, that can be used to compute efficiently local curvature tensors on triangulated surfaces.

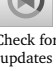

## 1   Introduction

In the last decades, biophysics and soft matter physics have provided many examples where the geometry of the system plays a major role to understand the structural and mechanical properties of materials such as colloids, liquid crystals, membranes, cells and tissues [1–6]. In the case of a fluid membrane for instance, the local mean and Gaussian curvatures of the surface are crucial ingredients of its energy functional [1,7]. The description of membranes that exhibit a polar (or nematic) order requires not only the knowledge of the principal curvatures, but that of the full local curvature tensor, as couplings between the polar order parameter and the curvature tensor are allowed by symmetry [8,9]. Moreover, active membranes such as the cell cortex or cell monolayers can exert active tensions and active moments which generically couple to the local curvature tensor [10]. Such couplings can for instance be a consequence of a preferential alignment of filaments of the cytoskeleton along the axis of largest curvature. At the macroscopic level, such systems can be described using tools from (nonequilibrium) statistical mechanics [10] and from differential geometry [11], and the existence of a smooth manifold and a well-defined curvature tensor is assumed. It is therefore crucial to understand how the macroscopic continuum description in terms of local metric and local curvature tensors can be obtained from the coarse-graining of discrete objects such as molecules, colloids or cells.

Surfaces represented by discrete triangles or polygons are ubiquitous as they arise from the discretization of a smooth surface for numerical purposes, or from experimental data on real geometries. Defining curvature measures on such polygonal or triangulated surfaces is a long-standing problem which has been addressed extensively in mathematics [12–14]. Despite these important contributions, that we discuss more below, a robust and practical definition of a local curvature tensor that converges to the continuum curvature tensor in the limit of infinitesimal triangulation remains however elusive for such surfaces. Several methods have been proposed to compute curvature tensors from triangulated surfaces. Some of them rely on a local fit of the smooth surface to the triangulated manifold [15,16]. The resulting curvature tensor depends on the method used and such approaches are unsatisfactory in cases where the surface is intrinsically made of triangles or polygons, as cellular tissues or foams, for instance. Other methods are directly built on the triangulated surface, but their definitions rely on arbitrary choices, leading to a proliferation of algorithms [17–20].

The development of integral geometry [12–14] has however provided physicists with new tools to quantify shapes and curvature of surfaces, for instance by using Minkowski functionals [21,22]. Minkowski functionals are defined locally on open or closed surfaces and can take scalar or tensorial values [12–14]. They possess several properties that make them especially appealing: they are additive, continuous, motion covariant (invariant for the scalars) and span the space of scalar and tensor-valued valuations on convex shapes [23–27]. The last property implies in particular that any extensive and motion invariant scalar functional can be written as a linear combination of the Minkowski scalars [21]. For a closed body embedded in a three-dimensional space, there are only $d + 1 = 4$ Minkowski scalars: the volume of this body, its surface area, its integrated mean curvature and its integrated Gaussian curvature (or Euler characteristic). Minkowski tensors are a generalization of the Minkowski scalars to ten-

sorial quantities [22, 28] and share their appealing properties. For the reasons stated above, the Minkowski scalars have proven extremely robust to evaluate the local curvature of surfaces [29, 30], while the rank-2 Minkowski tensors have been used to analyze anisotropy in a wide range of phenomena ranging from the shape of neuronal cells in the brain [31] to the shape of galaxies in the universe [32]. Their robustness also permits to use them on cellular and discretized structures, and a coarse-grained evaluation of Minkowski tensors can even be obtained for pixelated images and polygonal surfaces defined on grids [22]. Higher-rank Minkowski tensors have also been used recently for shape reconstruction [33, 34], relying on the fact that a polytope in dimension $d$ with $m$ facets is uniquely determined by its surface tensors up to rank $m-d+2$ [34]. Finally, we note that curvature-weighted tensorial measures have also been used with success in density functional theories for fluids made of arbitrarily-shaped convex hard particles [35, 36].

In this paper, inspired by these concepts from integral geometry, we provide a definition for a local integrated curvature tensor. This integrated curvature tensor is defined as the sum of two local Minkowski tensors. It is well-defined on polygonal surfaces and it is additive in the sense of Minkowski functionals. Because of this property, we can use this tensor to define a coarse-grained curvature tensor. We show that this coarse-grained curvature tensor on arbitrary triangulated surfaces displays robust convergence to the curvature tensor of the corresponding smooth surface.

The paper is organized as follows. We first define an integrated curvature tensor for a smooth surface. Using a parallel body construction, we then translate this definition to a triangulated surface, and discuss the properties of this triangle-based curvature tensor. In Sec. 3, we discuss the convergence properties of this definition on a set of simple smooth surfaces. We then present an algorithm to compute this curvature tensor on any triangulated surface by extending the algorithm of Ref. [22]. We illustrate this procedure on a few surfaces and compute principal curvatures and directions of principal curvature of the coarse-grained curvature tensor for these surfaces.

## 2 Integrated curvature tensor

### 2.1 Differential geometry of curved surfaces

We first consider the case of a smooth surface that can be described using differential geometry. Let $\mathcal{S}$ be a two-dimensional surface parameterized by two coordinates and let $\mathbf{x}(s^1, s^2) \in \mathcal{S}$ be a point on this surface. At each point on the surface we associate a local basis composed of two tangent vectors $\mathbf{e}_1$ and $\mathbf{e}_2$, and a normal vector $\mathbf{n}$, defined as:

$$\mathbf{e}_1 = \frac{\partial \mathbf{x}}{\partial s^1}, \quad \mathbf{e}_2 = \frac{\partial \mathbf{x}}{\partial s^2}, \quad \mathbf{n} = \frac{\mathbf{e}_1 \times \mathbf{e}_2}{|\mathbf{e}_1 \times \mathbf{e}_2|}, \tag{1}$$

see Fig. 1 for illustration. The vectors $\mathbf{e}_i$ form the covariant base of the tangent space. We furthermore define the contravariant base $\mathbf{e}^j$ as $\mathbf{e}_i \cdot \mathbf{e}^j = \delta_i^j$, where $\cdot$ denotes the scalar product, indices run from 1 to 2, and $\delta_i^j$ is the Kronecker symbol. We also define the local metric tensor with components $g_{ij} = \mathbf{e}_i \cdot \mathbf{e}_j$ in the local basis $\mathbf{e}_i$. Indices can be raised or lowered by contraction with the metric tensor: $a^i = g^{ij} a_j$ where a summation over repeated indices is implied here and in the following. We denote by $d\ell$ a line element on the surface with $d\ell^2 = g_{ij} ds^i ds^j$, and $dA = \sqrt{g} ds^1 ds^2$ the components of an area element, where $g = \det g_{ij}$ is the determinant of the metric tensor. We finally define the local curvature tensor with components $c_{ij}$ given by:

$$c_{ij} = -(\partial_i \partial_j \mathbf{x}) \cdot \mathbf{n}, \tag{2}$$

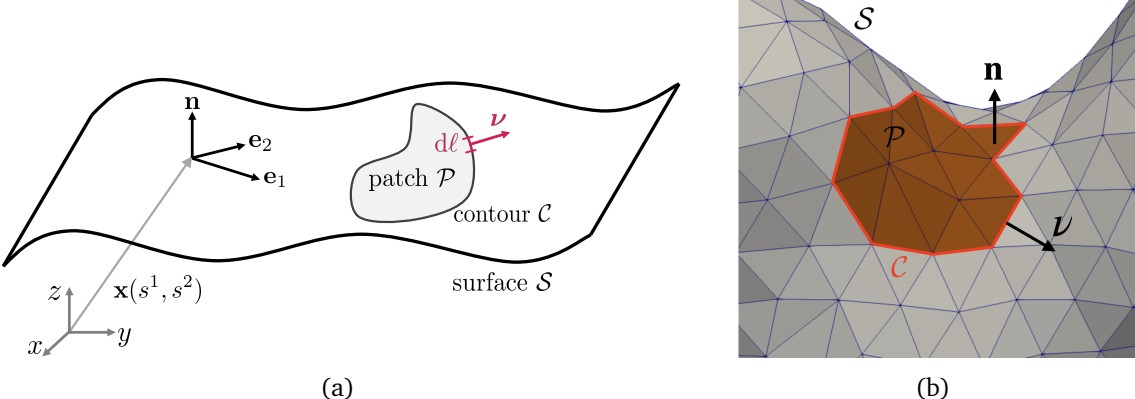

Figure 1: Notation used for a smooth surface **(a)** and a triangulated surface **(b)**. For the triangulated surface, the normal vector **n** is defined everywhere on the surface (including on the sharp edges between triangles) using a parallel body construction (see text).

where $\partial_i = \partial/\partial s^i$. The coordinate-independent form $\boldsymbol{c}$ of the curvature tensor is written as:

$$\boldsymbol{c} = c_{ij}\mathbf{e}^i \odot \mathbf{e}^j, \tag{3}$$

where $\boldsymbol{a} \odot \boldsymbol{b} \equiv (\boldsymbol{a} \otimes \boldsymbol{b} + \boldsymbol{b} \otimes \boldsymbol{a})/2$ is the symmetric tensor product. The curvature tensor describes changes in the local basis when moving on the surface as described by the Gauss–Weingarten equations [11]:

$$\nabla_j \mathbf{n} = c_{ij}\mathbf{e}^i, \tag{4a}$$

$$\nabla_j \mathbf{e}_i = -c_{ij}\mathbf{n}, \tag{4b}$$

where $\nabla_j$ denotes the covariant derivative [10, 11].

## 2.2 The integrated curvature tensor

Equation (3) is well-defined for smooth surfaces but difficult to define for surfaces built from discrete components such as particles, molecules, polygons or triangles. We therefore define a coarse-grained curvature measure defined by integrating the curvature tensor over a finite surface patch $\mathcal{P}$. This leads to the definition of the *integrated* curvature tensor $\boldsymbol{M}$:

$$\boldsymbol{M} = \int_{\mathcal{P}} dA\,\boldsymbol{c} = \int_{\mathcal{P}} dA\, c_{ij}\mathbf{e}^i \odot \mathbf{e}^j. \tag{5}$$

Using the Gauss–Weingarten equation (4a), the integrand of Eq. (5) can be rewritten as

$$c_{ij}\mathbf{e}^i \odot \mathbf{e}^j = \nabla_j \mathbf{n} \odot \mathbf{e}^j = \nabla_j(\mathbf{n} \odot \mathbf{e}^j) - \mathbf{n} \odot \nabla_j \mathbf{e}^j$$
$$= \nabla_j(\mathbf{n} \odot \mathbf{e}^j) + c_i^i \mathbf{n} \odot \mathbf{n},$$

where the second line has been obtained using Eq. (4b). Then, using the divergence theorem on a curved surface [10, 37], the integrated curvature tensor is rewritten as a sum of a boundary contribution and a surface term:

$$\boldsymbol{M} = \int_{\mathcal{P}} dA\, c_i^i \mathbf{n} \odot \mathbf{n} + \oint_{\mathcal{C}} d\ell\, \nu_i \mathbf{e}^i \odot \mathbf{n}. \tag{6}$$

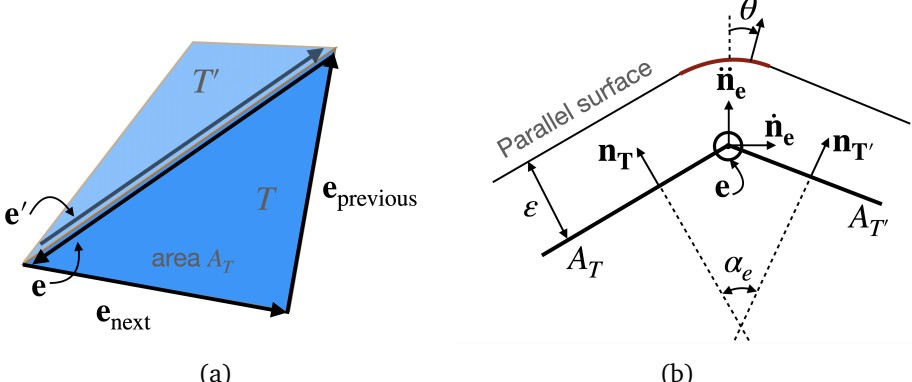

|      |      |
|:----:|:----:|
| (a)  | (b)  |

Figure 2: Definition and notation used for a triangulated surface and its parallel body, following Ref. [22]. **(a)** Definition of the edges using a doubly connected edge list data structure. Each edge **e** is oriented and uniquely assigned to a triangle $T$, and has a counter-oriented edge $\mathbf{e}'$ assigned to the adjacent triangle $T'$. An oriented edge is also unambiguously assigned to the previous edge $\mathbf{e}_{\text{previous}}$ and to the next edge $\mathbf{e}_{\text{next}}$. **(b)** Cross-sectional view along the oriented edge **e**. The parallel surface of width $\varepsilon$ regularizes the triangulated surface and allows the computation of integrated surface quantities. See main text for the definitions of the quantities.

Here, $\mathcal{C}$ denotes a contour enclosing the surface patch $\mathcal{P}$, and $\boldsymbol{\nu} = \nu_i \mathbf{e}^i$ is a unit vector, tangent to $\mathcal{P}$, outward-pointing and normal to the contour $\mathcal{C}$ (see Fig. 1).

The definition (6) of the integrated curvature tensor has the form of a sum of two Minkowski tensors. The surface term (first term in Eq. (6)) is proportional to the local three-dimensional Minkowski tensor $W_2^{0,2}$ [22]. The boundary term (second term in Eq. (6)), is formally a non-Euclidean two-dimensional Minkowski tensor embedded in three-dimensional space. The representation of the integrated curvature tensor given in Eq. (6) reduces for a closed surface $\mathcal{S}$ to a single rank-2 Minkowski tensor $W_2^{0,2} = (1/6) \oint_{\mathcal{S}} dA\, c_i^i \mathbf{n} \odot \mathbf{n}$ [22], since the contour term vanishes in this case. Importantly, both terms in Eq. (6) are additive and well-defined for a triangulated surface, as we see in the next section.

Furthermore, the integrated mean curvature $M$ over the patch $\mathcal{P}$ can be directly obtained by taking the trace of the integrated curvature tensor defined by Eq. (6):

$$M = \frac{1}{2} \int_{\mathcal{P}} dA\, c_i^i = \frac{1}{2} \text{Tr}\, \boldsymbol{M}\,, \tag{7}$$

since $\text{Tr}\left(\mathbf{e}^i \odot \mathbf{n}\right) = 0$ and $\text{Tr}\left(\mathbf{n} \odot \mathbf{n}\right) = 1$.

Finally, the integrated curvature tensor can be normalized by the area $A = \int_{\mathcal{P}} dA$ of patch $\mathcal{P}$ to obtain the *coarse-grained* curvature tensor $\boldsymbol{C}$:

$$\boldsymbol{C} = \boldsymbol{M}/A, \tag{8}$$

which will be crucial to define a local curvature tensor on polygonal surfaces.

## 2.3 Integrated curvature tensor on triangulated surfaces

In the following, we consider triangulated surfaces for simplicity. Note that our formalism can be applied to any polygonal surface. The parallel body construction of a triangulated surface, that we describe below, will allow us to define the integrated curvature tensor of surface patches represented by planar triangles and even for a single triangle by using Eq. (6).

For this purpose, we first define some notation following Ref. [22]. On the triangulated surface, each edge $\mathbf{e}$ is oriented and uniquely assigned to a triangle $T$, and has a counter-oriented edge $\mathbf{e}'$ assigned to the adjacent triangle $T'$ (see Fig. 2). An oriented edge is also unambiguously assigned to the previous edge $\mathbf{e}_{\text{previous}}$ and the next edge $\mathbf{e}_{\text{next}}$. This description corresponds to a doubly connected edge list data structure. The normal vector on a triangle $T$ is defined as $\mathbf{n}_T = (\mathbf{e}_{\text{previous}} \times \mathbf{e})/|\mathbf{e}_{\text{previous}} \times \mathbf{e}|$.

The normal vectors $\mathbf{n}_T$ and $\mathbf{n}_{T'}$ of triangles $T$ and $T'$ span the angle $\alpha_{\mathbf{e}} \in (-\pi, \pi]$ at their common edge $\mathbf{e}$. Along each edge $\mathbf{e}$, we define the normalized edge vector $\hat{\mathbf{e}}$, the mean normal vector $\ddot{\mathbf{n}}_{\mathbf{e}}$ and its normal $\dot{\mathbf{n}}_{\mathbf{e}}$, such that

$$\hat{\mathbf{e}} = \frac{\mathbf{e}}{|\mathbf{e}|}, \quad \ddot{\mathbf{n}}_{\mathbf{e}} = \frac{\mathbf{n}_T + \mathbf{n}_{T'}}{|\mathbf{n}_T + \mathbf{n}_{T'}|}, \quad \dot{\mathbf{n}}_{\mathbf{e}} = \hat{\mathbf{e}} \times \ddot{\mathbf{n}}_{\mathbf{e}} \tag{9}$$

form an orthonormal local basis.

By inflating the triangulated surface by an infinitesimal width $\varepsilon$, we construct a parallel smooth surface that regularizes the sharp edges between triangles by portions of cylinders and portions of spheres which allow the computation of integrated surface quantities (see Fig. 2). We first focus on the computation of the integrated curvature tensor of a single triangle, as the integrated curvature of a patch of triangles is obtained by summing the contribution of individual triangles (see below). The integrated curvature tensor of a single triangle is defined by computing Eq. (6) over the triangle and over the portions of the infinitesimal cylinders that connect it to its neighbors and then by taking the limit $\varepsilon \to 0$. The area contribution appearing in Eq. (6) for a single triangle reads:

$$\int_{\mathcal{P}} dA\, \mathbf{n} \odot \mathbf{n} c_i^i = \lim_{\varepsilon \to 0} \sum_{\mathbf{e} \in T} \int_0^{|\mathbf{e}|} d\ell \int_{-\alpha_{\mathbf{e}}/2}^{\alpha_{\mathbf{e}}(a_{\mathbf{e}} - 1/2)} \varepsilon d\theta\, \mathbf{n} \odot \mathbf{n} \frac{1}{\varepsilon}, \tag{10}$$

where the sum runs over the edges of triangle $T$ and where $\mathbf{n}$ is the normal vector along the portion of cylinder and is parameterized by the angle $\theta$ such that $\mathbf{n} = \cos\theta\, \ddot{\mathbf{n}}_{\mathbf{e}} + \sin\theta\, \dot{\mathbf{n}}_{\mathbf{e}}$ (see App. A and Fig. 8 for details). Note that we integrate over a weighted portion of the cylinder and have therefore introduced the weight fraction $a_{\mathbf{e}}$, which we discuss in the following. Similarly, the boundary contribution is given by:

$$\oint_{\mathcal{C}} d\ell\, v_i \mathbf{e}^i \odot \mathbf{n} = \lim_{\varepsilon \to 0} \sum_{\mathbf{e} \in T} \int_0^{|\mathbf{e}|} d\ell\, \boldsymbol{v}(\theta_{\mathrm{f}}) \odot \mathbf{n}(\theta_{\mathrm{f}}), \tag{11}$$

where $\theta_{\mathrm{f}} = \alpha_{\mathbf{e}}(a_{\mathbf{e}} - 1/2)$ is the final angle of the portion of cylinder in the local basis defined by Eq. (9), and $\boldsymbol{v}(\theta_{\mathrm{f}}) = \mathbf{n}(\theta_{\mathrm{f}} + \pi/2) = \hat{\mathbf{e}} \times \mathbf{n}(\theta_{\mathrm{f}})$ (see Fig. 8).

Evaluating and summing the contributions from Eqs. (10) and (11), we obtain the *integrated* curvature tensor $\boldsymbol{M}_T$ of a single triangle as:

$$\begin{aligned} M_T = \frac{1}{4} \sum_{\mathbf{e} \in T} |\mathbf{e}| \Big[ &(2a_{\mathbf{e}}\alpha_{\mathbf{e}} + \sin\alpha_{\mathbf{e}} + \sin(\alpha_{\mathbf{e}} - 2a_{\mathbf{e}}\alpha_{\mathbf{e}}))\ddot{\mathbf{n}}_{\mathbf{e}} \odot \ddot{\mathbf{n}}_{\mathbf{e}} \\ &+ (2a_{\mathbf{e}}\alpha_{\mathbf{e}} - \sin\alpha_{\mathbf{e}} - \sin(\alpha_{\mathbf{e}} - 2a_{\mathbf{e}}\alpha_{\mathbf{e}}))\dot{\mathbf{n}}_{\mathbf{e}} \odot \dot{\mathbf{n}}_{\mathbf{e}} \\ &+ 4\cos(a_{\mathbf{e}}\alpha_{\mathbf{e}})\cos(\alpha_{\mathbf{e}} - a_{\mathbf{e}}\alpha_{\mathbf{e}})\ddot{\mathbf{n}}_{\mathbf{e}} \odot \dot{\mathbf{n}}_{\mathbf{e}} \Big]. \end{aligned} \tag{12}$$

Note that for concave triangles patches, the sign of $\alpha_{\mathbf{e}}$ in Eq. (12) has to be changed (see App. A.3). The weight fraction $a_{\mathbf{e}}$ determines the fraction of the integrated curvature associated with bond $\mathbf{e}$ that is assigned to triangle $T$. Additivity of the integrated curvature tensor imposes $a_{\mathbf{e}} + a_{\mathbf{e}}' = 1$, where $a_{\mathbf{e}}'$ is the weight fraction associated to triangle $T'$.

The integrated curvature tensor has the following properties. First, the integrated curvature tensor $\boldsymbol{M}$ of a patch $P$ of triangles is directly obtained by summing their individual contributions:

$$\boldsymbol{M} = \sum_{T \in P} \boldsymbol{M}_T \, , \tag{13}$$

where the sum runs over the triangles $T$ forming patch $P$. Furthermore, similarly to Eq. (7) in the smooth case, the trace of the triangle-based integrated curvature tensor reads:

$$\text{Tr}\,\boldsymbol{M}_T = \sum_{\mathbf{e} \in T} |\mathbf{e}| a_{\mathbf{e}} \alpha_{\mathbf{e}} \, , \tag{14}$$

and corresponds to the weighted sum of the integrated mean curvatures of each edge. Note finally that $\boldsymbol{M}_T = 0$ if the triangle $T$ is surrounded by co-planar triangles (see App. A.2 for details).

## 2.4 Coarse-grained curvature tensor on triangulated surfaces

The integrated curvature tensor is a cumulative measure of curvature over a surface patch. It is often convenient and useful to define coarse-grained curvature measures, for example to define the curvature on a surface that is formed by discrete particles. Starting from individual triangles, we define the *coarse-grained* curvature tensor $\boldsymbol{C}_T$ of triangle $T$ as:

$$\boldsymbol{C}_T = \boldsymbol{M}_T / A_T \, , \tag{15}$$

where $A_T$ is the area of triangle $T$. Similarly, the coarse-grained curvature tensor for a patch $P$ is given by Eq. (8) with $\boldsymbol{M}$ defined in Eq. (13).

If a smooth surface is represented by a triangulation, we demand that the coarse-grained curvature tensor of a triangle converges to the curvature of the smooth surface in the limit where triangles become infinitesimal. This requirement fixes the value of the weight $a_{\mathbf{e}}$ as follows. The coarse-grained mean curvature $H_T = \sum_{\mathbf{e} \in T} H_{\mathbf{e}}$ of triangle $T$ combines contributions $H_{\mathbf{e}} = |\mathbf{e}| \alpha_{\mathbf{e}} a_{\mathbf{e}} / (2 A_T)$ from all edges. Convergence to the curvature tensor of a smooth surface requires that the mean curvature associated with a bond is the same for both adjacent triangles $T$ and $T'$. Thus, $H_{\mathbf{e}} = H'_{\mathbf{e}}$, which together with $a_{\mathbf{e}} + a'_{\mathbf{e}} = 1$ uniquely determines

$$a_{\mathbf{e}} = \frac{A_T}{A_T + A_{T'}} \, . \tag{16}$$

Equation (15), together with Eqs. (12) and (16), define the coarse-grained curvature tensor, which is the main result of the paper. This coarse-grained curvature tensor provides a robust definition of curvature on triangulated surfaces, defined for a single triangle and its direct neighbors. The definition (12) of the triangle-based integrated curvature tensor has several appealing properties: (i) owing to the parallel body construction, it is well-defined for triangulated surfaces, (ii) it is additive, and can therefore be used for coarse-graining using Eq. (13), (iii) it displays robust convergence when a triangulation approximates a smooth surface, as we show in the following section.

# 3 Approximation of smooth surfaces by triangulations

## 3.1 Convergence of the triangle-based integrated curvature tensor to a continuum limit

We have shown that the coarse-grained curvature tensor can be computed for a triangulated surface. We now show that it converges to the curvature tensor of a smooth surface in the

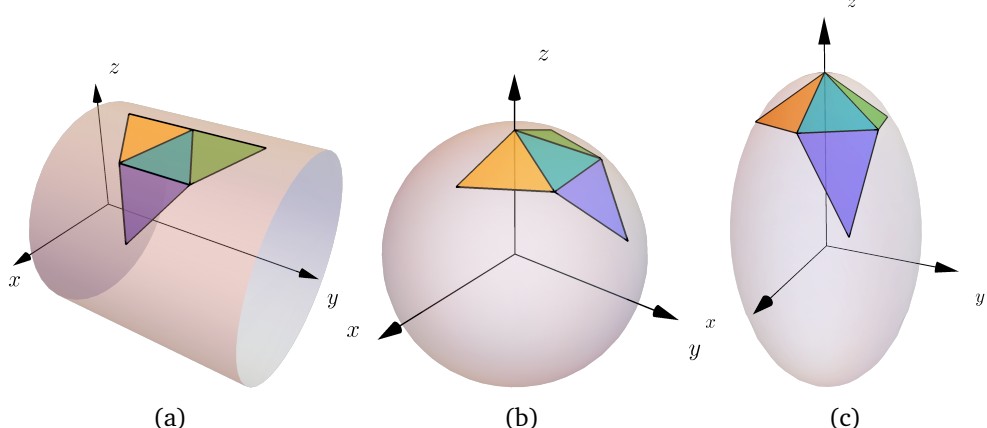

Figure 3: Patch of triangles on a cylinder **(a)**, on a sphere **(b)**, and on an ellipsoid **(c)**. The triangle-based curvature tensor $C_T$ is computed for the central blue triangle $T$, using curvature information stemming form its direct neighbors.

limit where the triangle size becomes small. To this end, we consider the curvature tensor of a smooth cylinder, sphere and ellipsoid. For a smooth surface, the curvature tensor (3) is a $3 \times 3$ matrix of rank 2, and has three eigenvalues. We denote by $\lambda_{1,2}$ the two principal curvatures, and by $\lambda_3 = 0$ the last eigenvalue, whose eigenvector is parallel to the surface normal. We compare the curvature tensor of smooth surfaces with the coarse-grained curvature tensor $C_T$ of a single triangle $T$ surrounded by its three direct neighbors with triangle vertices located on the surfaces (see Fig. 3). We determine the coarse-grained curvature tensor of the central triangle using Eqs. (15) and (12) with an area-weight (16). We denote by $\Lambda_{1,2}$ the principal curvatures of the discrete curvature tensor that corresponds to $\lambda_{1,2}$, and by $\Lambda_3$ the eigenvalue that converges to 0 and for which the associated eigenvector is parallel to the surface normal as the areas of the triangles become small.

**Cylinder.** We first consider an equilateral triangle of edge length $\ell$ surrounded by three equilateral triangles. All triangle vertices are located on the surface of a cylinder of radius $R$. For the simple case where three of the triangles are coplanars (see Fig. 3a), the eigenvalues read

$$
\begin{aligned}
\Lambda_1 &= \frac{\cos^{-1}\left(1 - \frac{3\varepsilon^2}{8}\right)}{\sqrt{3}R\varepsilon} + \frac{\sqrt{2 + 3\varepsilon^2}}{2\sqrt{2}R}, \\
\Lambda_2 &= \frac{\cos^{-1}\left(1 - \frac{3\varepsilon^2}{8}\right)}{\sqrt{3}R\varepsilon} - \frac{\sqrt{2 + 3\varepsilon^2}}{2\sqrt{2}R}, \\
\Lambda_3 &= 0,
\end{aligned}
\tag{17}
$$

where $\varepsilon = \ell/R$. To lowest order in $\varepsilon$ we have:

$$
\Lambda_1 = \frac{1}{R} + \frac{25\varepsilon^2}{64R} + \mathcal{O}\left(\varepsilon^3\right), \quad \Lambda_2 = -\frac{23\varepsilon^2}{64R} + \mathcal{O}\left(\varepsilon^3\right),
\tag{18}
$$

showing a convergence as $\varepsilon^2$ to the curvature tensor of a smooth cylinder with $\lambda_1 = 1/R$ and $\lambda_2 = 0$. An alternative tiling, where none of the triangles are co-planar, is provided in App. A.3. In this case, we observe a slower convergence, linear in $\varepsilon$, to the same eigenvalues.

**Sphere.** Considering four equilateral triangles on a sphere of radius $R$, we obtain the eigenvalues of the discrete curvature tensor $C_T$, see Fig. 3b. At lowest order in $\varepsilon = \ell/R$, we obtain:

$$
\Lambda_1 = \frac{1}{R} + \frac{5\varepsilon^2}{18R} + \mathcal{O}\left(\varepsilon^3\right), \quad \Lambda_2 = \frac{1}{R} + \frac{5\varepsilon^2}{18R} + \mathcal{O}\left(\varepsilon^3\right), \quad \Lambda_3 = -\frac{5\varepsilon^2}{18R} + \mathcal{O}\left(\varepsilon^3\right),
\tag{19}
$$

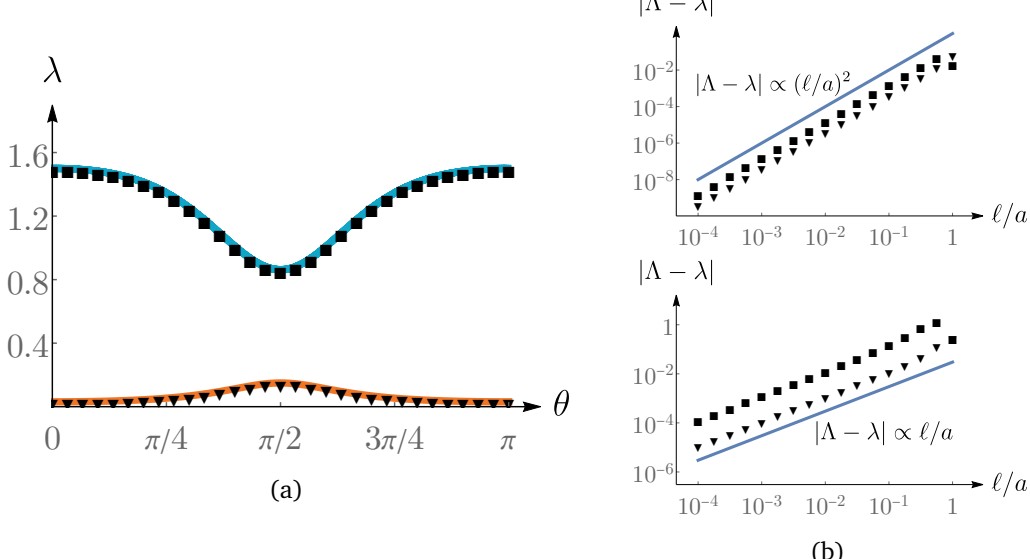

Figure 4: Convergence of the triangle-based curvature tensor on an ellipsoid with major axes $a = 1, b = 7, c = 3/2$. **(a)** Polar angle dependence of the principal curvatures of the smooth curvature tensor (solid lines) and of the discrete one (black symbols). The triangle-based curvature tensor has been computed for $\varphi = \pi/3$ and $\ell/a = 10^{-3}$. **(b)** Convergence of the triangle-based principal curvatures $\Lambda_{1,2}$ to the smooth principal curvatures $\lambda_{1,2}$ for a patch of triangles located at $\theta = 0$, $\varphi = \pi/3$ (top) and $\theta = \pi/3$, $\varphi = \pi/3$ (bottom).

showing a convergence as $\varepsilon^2$ to the curvature tensor of a smooth sphere with $\lambda_{1,2} = 1/R$.

**Ellipsoid.** We now consider an ellipsoid defined by the equation

$$x = a\sin\theta\cos\varphi\,, \quad y = b\sin\theta\sin\varphi\,, \quad z = c\cos\theta\,. \tag{20}$$

For simplicity, we generate a triangular patch on the ellipsoid by using four equilateral triangles on the unit sphere and performing an affine transformation of the sphere by rescaling the $x, y, z$ coordinates by the factors $a, b, c$, respectively (see Fig. 3c). The curvature tensor on the ellipsoid is a function of the polar and azimuthal angles $\theta$ and $\varphi$. In Fig. 4a, we show the comparison between the principal curvatures of the discrete and of the smooth curvature tensor on an ellipsoid with major axes $a = 1, b = 7, c = 3/2$. We consider a patch of triangles with a vertex of the central triangle located at $\theta = 0, \varphi = \pi/3$. In this case, the principal curvatures converge as $\varepsilon^2$ for small $\varepsilon = \ell/a$ to the principal curvatures on the ellipsoid (see top panel of Fig. 4b). The convergence is linear in $\varepsilon$ at $\theta = \pi/3, \varphi = \pi/3$ (bottom panel of Fig. 4b). This difference in convergence can be explained by the fact that for $\theta = 0, \varphi = \pi/3$ all triangles are equal in shape, while at $\theta = \pi/3, \varphi = \pi/3$ the symmetry between triangles is broken. Note that the third eigenvalue $\Lambda_3$ converges to 0 as $\varepsilon^2$ in both cases.

## 3.2 Directions of principal curvature on triangulated surfaces

The definition (15) can be used to obtain the local principal directions of the coarse-grained curvature tensor on triangulated surfaces. For this purpose, we have extended the karambola package designed in Ref. [22] to compute the triangle-based curvature tensor on arbitrary surfaces. We provide the corresponding C++ code in App. B. As an illustration, we have determined the principal curvatures and principal directions of curvature of the triangle-based

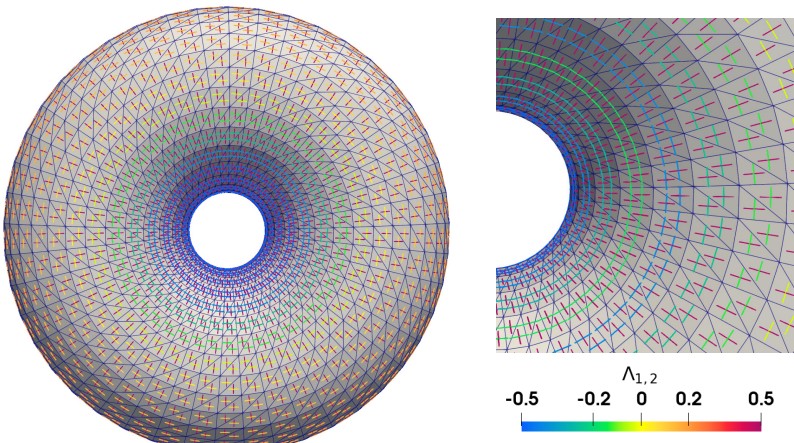

Figure 5: Principal curvatures and principal directions of curvature of the triangle-based curvature tensor for a Clifford torus (triangulated by 2880 triangles).

curvature tensor for three examples of closed surfaces with different topology, see Figs. 5, 6 and 7.

Figure 5 displays the principal curvatures and principal directions of curvature for a triangulated Clifford torus. The Clifford torus is an axisymmetric torus with circular cross-section and a ratio $\sqrt{2}$ of the radii of its two generating circles [38]. It has the important property to minimize $G = \int_{\mathcal{S}} dA (c_i^i)^2$ for toroidal topology. We indicate on each triangle the principal directions of the triangle-based curvature tensors as bars with a color which indicates the magnitude of the associated principal curvature. Note that one curvature is constant and positive, while the other is positive (red) on the outer part of the torus and negative (blue) on the inner part of the torus.

From this Clifford torus, we perform a special conformal transformation, which consists of an inversion $\mathbf{R}' = \mathbf{R}/R^2$, followed by a translation by a vector $\mathbf{t}$ and a second inversion, such that a point $\mathbf{R}$ is mapped to [30]:

$$\mathbf{R}' = \frac{\mathbf{R}/R^2 - \mathbf{t}}{(\mathbf{R}/R^2 - \mathbf{t})^2} \, . \tag{21}$$

For $\mathbf{t} = 1.25 r \, \mathbf{e}_x$, with $r$ the larger radius of the Clifford torus and $\mathbf{e}_x$ a unit vector orthogonal to the axis of symmetry of the torus, the transformation (21) applied to the Clifford torus yields a non-axisymmetric torus with the same minimal value of $G$ (see Fig. 6). Indicated are again the principal curvatures and principal directions of curvature of the triangle-based curvature tensor for each triangle.

The third example, displayed in Fig. 7, is the triangulated Lawson surface of genus 2 [30, 39]. The Lawson surface minimizes $G$ for genus-2 surfaces. The principal curvatures and principal directions of curvature of the triangle-based curvature tensor are plotted on each triangle. These examples show that the coarse-grained curvature tensor, defined on a triangulation, provides a good approximation for the curvature tensor of the underlying smooth surfaces.

## Conclusion

In this paper, starting from the definition of the integrated curvature tensor for a smooth surface, we have used a parallel body construction to define an integrated curvature tensor for

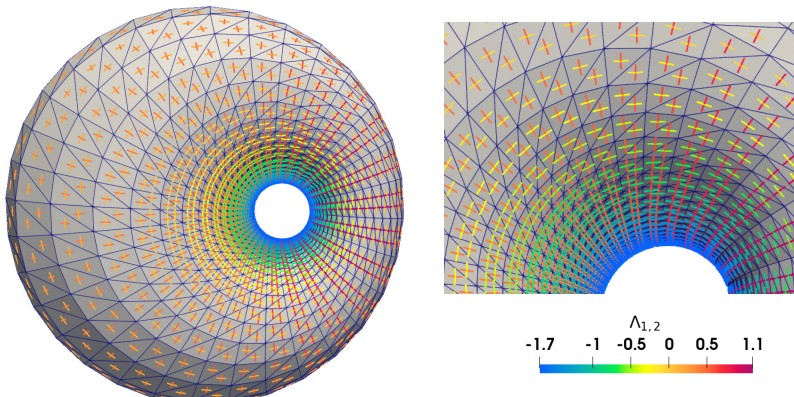

Figure 6: Principal curvatures and principal directions of curvature of the triangle-based curvature tensor for a Clifford torus after a special conformal transformation defined in Eq. (21) (triangulated by 2880 triangles).

polygonal and in particular for triangulated surfaces. Parallel body construction is often used in the context of integral geometry, and is at the heart of the definition and evaluation of Minkowski functionals on polygonal manifolds [12–14, 22]. Our definition of the integrated curvature tensor involves a sum of two Minkowski tensors (see Eq. (6)). The first tensor is the surface contribution that is weighted by the mean curvature of the surface patch on which it is evaluated and is commonly used as a local curvature measure on its own. The second tensor is a boundary contribution defined on the contour of the patch, and thus corresponds to a two-dimensional Minkowski tensor embedded in three dimensional space. Such an embedding of a Minkowski tensor to a higher dimensional space appears naturally in our formalism and provides an extension of Minkowski tensors to non-Euclidean geometries. Adding the surface and the boundary contributions together provides the information to construct a local curvature tensor. Our approach illustrates the importance of the boundary contribution to obtain a robust convergence of the coarse-grained curvature tensor to that of smooth surfaces. We note that to define the integrated curvature tensor, it is not necessary to specify the value of the weight factor $a_e$. However, to define a coarse-grained curvature tensor, the weight factor takes a unique value given by Eq. (16).

Importantly, our approach provides a definition of curvature on non-smooth surfaces. This is relevant to build the continuum limit of the statistical physics of curved manifolds starting from structures made of particles or molecules. For example, in a fluid membrane molecular positions define triangular meshes from which the coarse-grained curvature tensor introduced here can be computed. Therefore, the differential geometry of a smooth surface can emerge from a structure composed of discrete objects in a continuum limit. The existence of such an explicit coarse-graining procedure is essential to make sense and to provide explicit computational tools for theories where couplings between the curvature tensor and an additional order parameter are allowed by symmetry. This is the case for instance for equilibrium fluid membranes that possess a nematic order parameter that can couple to the surface geometry [9, 40–42]. It is also true for nonequilibrium systems evolving on a curved manifold, where active couplings between a polar or nematic order parameter and the curvature tensor are generically allowed [10, 43, 44]. Importantly, our coarse-grained curvature tensor can also be used to compute the local curvature tensor of surfaces for which keeping track of the discreteness of its constituents is relevant. It is the case for instance of cell tissues. For such systems, triangle-based methods have been developed to decompose the deformations of flat tissues into cell contributions [45] and have contributed to understand the role of cellular processes in tissue patterning [46]. For curved tissues, such method is still lacking and

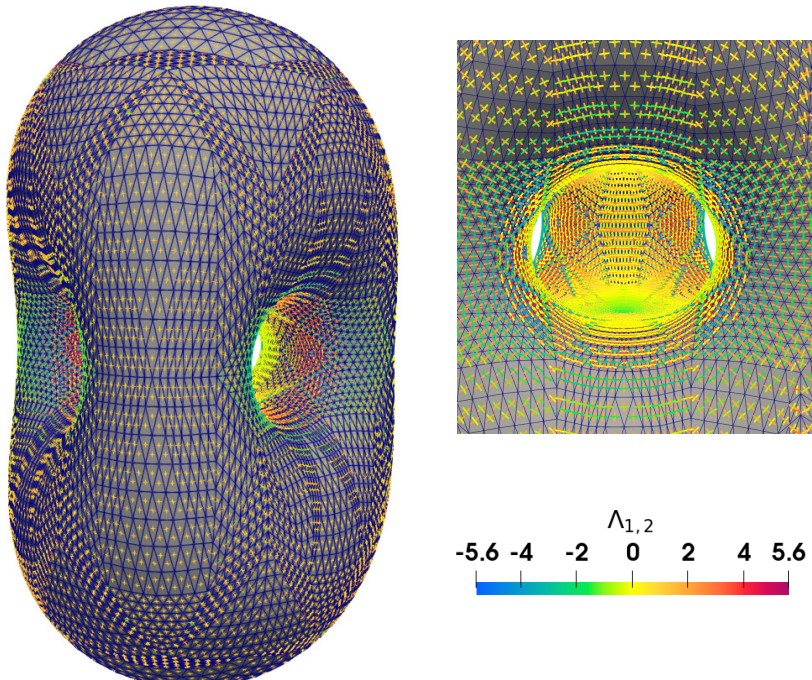

Figure 7: Principal curvatures and principal directions of curvature of the triangle-based curvature tensor for the Lawson surface with genus 2 (triangulated by 24576 triangles).

would require to also quantify changes of the curvature tensor as the tissue deforms. Our coarse-grained curvature tensor now provides the tool to perform such quantification.

Finally, a fast convergence of the coarse-grained curvature tensor to the smooth curvature tensor has been demonstrated for various surfaces. Using the karambola package designed in Ref. [22] and the code described in App. B, the coarse-grained curvature tensor of a triangulated surface can be computed efficiently. Importantly, the principal directions of curvature are computed with accuracy, as we have illustrated on Figs. 5, 6 and 7. An efficient tool is thus available to quantify curvature on triangulated surfaces, that could be used in the context of numerical simulation or experimental data processing, and also for computer vision and image processing.

## A  Integrated curvature tensor

### A.1  Computation of the integrated curvature tensor for a single triangle and its neighbors

We recall here the definition of the integrated curvature tensor $\boldsymbol{M}$ for a smooth surface as defined in Eq. (6):

$$\boldsymbol{M} = \int_{\mathcal{P}} \mathrm{d}A\, c_i^i \mathbf{n} \odot \mathbf{n} + \oint_{\mathcal{C}} \mathrm{d}\ell\, \nu_i \mathbf{e}^i \odot \mathbf{n}. \tag{22}$$

For a triangulated surface, one can compute the integrated curvature tensor using a parallel body construction as discussed in the main text. One can then first compute the surface integral term in Eq. (22). Since this term is weighted by the mean curvature, the flat part of the triangle

with zero curvature does not contribute. The portions of sphere that regularize the corners of the triangles yield a contribution of order $\varepsilon$ since their area is of order $\varepsilon^2$ while their mean curvature goes as $1/\varepsilon$. Therefore, contributions from triangle corners vanish, and only the infinitesimal portions of cylinder regularizing the edges have non-vanishing contributions in the limit $\varepsilon \to 0$, which read:

$$\int_{\mathcal{P}} dA \, \mathbf{n} \odot \mathbf{n} c_i^i = \lim_{\varepsilon \to 0} \sum_{\mathbf{e} \in T} \int_0^{|\mathbf{e}|} d\ell \int_{-\alpha_{\mathbf{e}}/2}^{\alpha_{\mathbf{e}}(a_{\mathbf{e}} - 1/2)} \varepsilon d\theta \, \mathbf{n} \odot \mathbf{n} \frac{1}{\varepsilon} , \tag{23}$$

where the sum runs over the edges of triangle $T$. Note that we integrate over a weighted portion of the cylinder and have therefore introduced the weighting factor $a_{\mathbf{e}}$.

The vector $\mathbf{n}$ is the normal vector along the portion of cylinder and is then parameterized by the angle $\theta$ such that $\mathbf{n} = \cos \theta \, \ddot{\mathbf{n}}_{\mathbf{e}} + \sin \theta \, \dot{\mathbf{n}}_{\mathbf{e}}$ (see Fig. 8). The surface integral thus reads:

$$\begin{aligned}
\int_{\mathcal{P}} dA \, c_i^i \mathbf{n} \odot \mathbf{n} = \frac{1}{4} \sum_{\mathbf{e} \in T} |\mathbf{e}| \Big[ &(2a_{\mathbf{e}} \alpha_{\mathbf{e}} + \sin \alpha_{\mathbf{e}} - \sin(\alpha_{\mathbf{e}} - 2a_{\mathbf{e}} \alpha_{\mathbf{e}})) \ddot{\mathbf{n}}_{\mathbf{e}} \odot \ddot{\mathbf{n}}_{\mathbf{e}} \\
&+ (2a_{\mathbf{e}} \alpha_{\mathbf{e}} - \sin \alpha_{\mathbf{e}} + \sin(\alpha_{\mathbf{e}} - 2a_{\mathbf{e}} \alpha_{\mathbf{e}})) \dot{\mathbf{n}}_{\mathbf{e}} \odot \dot{\mathbf{n}}_{\mathbf{e}} \\
&- 4 \sin(a_{\mathbf{e}} \alpha_{\mathbf{e}}) \sin(\alpha_{\mathbf{e}} - a_{\mathbf{e}} \alpha_{\mathbf{e}}) \ddot{\mathbf{n}}_{\mathbf{e}} \odot \dot{\mathbf{n}}_{\mathbf{e}} \Big] .
\end{aligned} \tag{24}$$

Second, the contour integral in Eq. (6) is evaluated by noting that the unit vector $\boldsymbol{\nu}$, tangent to the regularized triangle and normal to its contour lies at the end of the portions of cylinder, such that:

$$\oint_{\mathcal{C}} d\ell \, \nu_i \mathbf{e}^i \odot \mathbf{n} = \lim_{\varepsilon \to 0} \sum_{\mathbf{e} \in T} \int_0^{|\mathbf{e}|} d\ell \, \boldsymbol{\nu}(\theta_{\mathrm{f}}) \odot \mathbf{n}(\theta_{\mathrm{f}}) , \tag{25}$$

where $\theta_{\mathrm{f}} = \alpha_{\mathbf{e}}(a_{\mathbf{e}} - 1/2)$ is the final angle of the portion of cylinder in the local basis defined by Eq. (9), and $\boldsymbol{\nu}(\theta_{\mathrm{f}}) = \mathbf{n}(\theta_{\mathrm{f}} + \pi/2) = \hat{\mathbf{e}} \times \mathbf{n}(\theta_{\mathrm{f}})$ (see Fig. 8). Note that the portions of sphere that regularize the corners of the triangle do not contribute to the previous equation as the contour length on each portion of sphere is of order $\varepsilon$ and vanishes in the limit $\varepsilon \to 0$. We thus obtain for the contour term:

$$\oint_{\mathcal{C}} d\ell \, \nu_i \mathbf{e}^i \odot \mathbf{n} = \frac{1}{2} \sum_{\mathbf{e} \in T} |\mathbf{e}| \Big[ \sin(\alpha_{\mathbf{e}} - 2a_{\mathbf{e}} \alpha_{\mathbf{e}})(\ddot{\mathbf{n}}_{\mathbf{e}} \odot \ddot{\mathbf{n}}_{\mathbf{e}} - \dot{\mathbf{n}}_{\mathbf{e}} \odot \dot{\mathbf{n}}_{\mathbf{e}}) + 2 \cos(\alpha_{\mathbf{e}} - 2a_{\mathbf{e}} \alpha_{\mathbf{e}}) \ddot{\mathbf{n}}_{\mathbf{e}} \odot \dot{\mathbf{n}}_{\mathbf{e}} \Big] . \tag{26}$$

Summing Eqs. (24) and (26) yields Eq. (12) in the main text. For a patch $P$ of triangles, it is clear from Eqs. (23) and (25) that the integrated curvature tensor $M$ of the patch is given by adding individual triangles contribution, and yields Eq. (13) in the main text.

## A.2 Integrated curvature tensor for co-planar triangles

As expected, the integrated curvature tensor given by Eq. (12) vanishes for a triangle $T$ surrounded by co-planar triangles. This fact is however not obvious from Eq. (12) and we give details here. The first two terms in the sum appearing in Eq. (12) vanish since $\alpha_{\mathbf{e}} = 0$ for all edges for co-planar triangles. In addition, $\ddot{\mathbf{n}}_{\mathbf{e}} = \mathbf{n}_T$ is the triangle normal and is the same for all edges, and $\dot{\mathbf{n}}_{\mathbf{e}} = \boldsymbol{\nu}_{\mathbf{e}}$ is the unit vector normal to each edge in the plane of the triangle. The last term in Eq. (12) thus reduces to $\mathbf{n}_T \odot \sum_{\mathbf{e} \in T} |\mathbf{e}| \boldsymbol{\nu}_{\mathbf{e}}$. One can show, for instance by taking the scalar product with $\boldsymbol{\nu}_{\mathbf{e}'}$ and using the law of cosines, that $\sum_{\mathbf{e} \in T} |\mathbf{e}| \boldsymbol{\nu}_{\mathbf{e}}$ vanishes for any triangle, This shows that the integrated curvature tensor vanishes for a triangle surrounded by co-planar triangles.

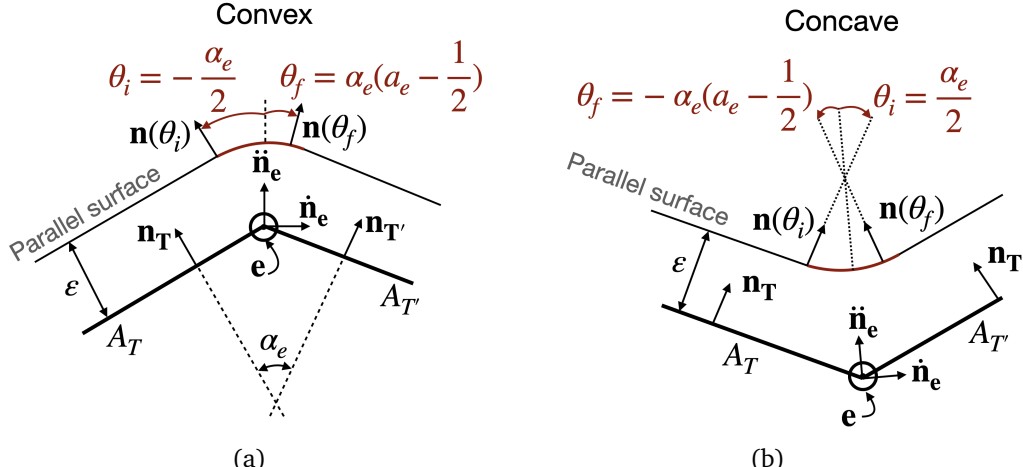

(a)                                                      (b)

Figure 8: Cross-sectional view of neighboring triangles in a convex (a), and concave (b) configurations. The angles $\theta_i$ and $\theta_f$ denote the lower and upper limits of integration, respectively.

### A.3   Concave decomposition

The derivation of Eq. (12) assumed a local convexity of the triangulated surface. As illustrated in Fig. 8, the generalization to concave edges is however straightforward as it simply requires to change the sign of $\alpha_{\mathbf{e}}$ for concave edges.

A simple illustration of this procedure occurs when considering a cylinder with triangles disposed in the direction orthogonal to the cylinder axis, see Fig. 9. In this case, the central triangle and the purple triangle form a concave edge for which the procedure highlighted above must be applied.

Similarly to the parallel tiling of the cylinder presented in the main text, we can obtain the eigenvalues of the coarse-grained curvature tensor and expand them in series of $\varepsilon = \ell/R$ to obtain:

$$\Lambda_1 = \frac{1}{R} + \frac{53\varepsilon^2}{288R} + \mathcal{O}(\varepsilon^3), \quad \Lambda_2 = \frac{\varepsilon}{8\sqrt{3}R} - \frac{7\varepsilon^2}{96R} + \mathcal{O}(\varepsilon^3), \quad \Lambda_3 = -\frac{\varepsilon}{8\sqrt{3}R} - \frac{7\varepsilon^2}{96R} + \mathcal{O}(\varepsilon^3).$$
(27)

Note that the convergence of the second and third eigenvalues is only linear in $\varepsilon$, while it is quadratic in the case of a parallel tiling (see Eq.(17)).

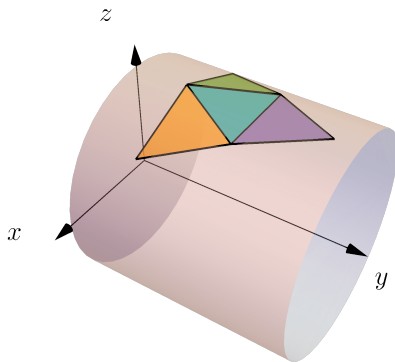

Figure 9: Patch of triangles on a cylinder. Tiling is orthogonal to the axis of the cylinder.

# B C++ code for computing the triangle-based integrated curvature tensor

We have extended the karambola package (https://morphometry.org/software/karambola/) to include the computation of the coarse-grained curvature tensor, its eigenvalues and eigenvectors. We provide below the C++ function that computes the *integrated* curvature tensor on a triangulated surface using Eq. (12). Similarly to the computation of other quantities that can be determined using the karambola package, the integrated curvature tensor can also be computed for labeled patches on the surface. If all triangles are labeled uniquely, the integrated curvature tensor is computed for each triangle. Note that in order to obtain Figs. 5, 6 and 7, we have computed the integrated curvature tensor of each triangle and normalized it by the triangle area to obtain the coarse-grained curvature tensor of each triangle.

```cpp
CompWiseMatrixMinkValResultType calculate_curvature(const Triangulation& surface){
//calculate c_int (integrated curvature)
CompWiseMatrixMinkValResultType c_int;
for (unsigned int l = 0; l<surface.n_triangles();l++){
  double A_t = surface.area_of_triangle(l);
    for (unsigned int k = 0; k<3;k++){
      if(surface.ith_neighbour_of_triangle(l,k) != NEIGHBOUR_UNASSIGNED){
        double A_tp= surface.area_of_triangle(surface.ith_neighbour_of_triangle(l,k));
        double ar= A_t/(A_t+A_tp);
        Vector n2=surface.normal_vector_of_triangle(surface.ith_neighbour_of_triangle(l,k));
        Vector n1= surface.normal_vector_of_triangle(l);
        double c= dot(n1,n2)/(norm(n1)*norm(n2));
        assert (c <= 1.000001);
        if (c>= 1) c=1.;
        double alpha= acos(c);

        Vector com_l = surface.com_of_triangle(l); //center of mass of triangle
        Vector com_k = surface.com_of_triangle(surface.ith_neighbour_of_triangle(l,k));
        Vector convex=com_l+n1-(com_k+n2);
        Vector concave=com_l-n1-(com_k-n2);
        if(norm(convex) < norm(concave)) alpha=-alpha;

        double e_norm= surface.get_edge_length(l,k);
        Vector e_c1= surface.get_pos_of_vertex(surface.ith_vertex_of_triangle(l,k));
        Vector e_c2= surface.get_pos_of_vertex(surface.ith_vertex_of_triangle(l,(k+1)%3));

        Vector e= (e_c2 - e_c1)/e_norm;
        Vector n_a= (n1 + n2)/(norm(n1+n2));
        Vector n_i= cross_product(e,n_a);

        double local_value= 0;
        for (unsigned int i = 0; i<3;i++){
          for (unsigned int j = 0; j<=i;j++){
          double term1=(2*ar*alpha+sin(alpha)+sin(alpha-2*ar*alpha)) * n_a[i]*n_a[j];
          double term2=(2*ar*alpha-sin(alpha)-sin(alpha-2*ar*alpha)) * n_i[i]*n_i[j];
          double term3=2*cos(ar*alpha)*cos(alpha-ar*alpha)*(n_a[i]*n_i[j]+n_a[j]*n_i[i]);
          local_value= e_norm*( term1 + term2 + term3 );
          c_int[surface.label_of_triangle(l)].result_(i,j) += 1/4.*local_value;}
        }
      }
    }
}
return c_int;
}
```

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
