# Peer review of "Coarse-grained curvature tensor on polygonal surfaces"

_SciPost Physics Core, doi:SciPost Phys. Core 5, 011 (2022)_

## Round 1 · Referee Report · Anonymous · 2021-6-8

Report

The authors use techniques from integral geometry to define a local coarse-grained curvature tensor on polygonal surfaces, so that one obtains in the continuum limit the correct principal curvatures and curvature directions. This is important for many numerical applications in soft matter physics where fluid interfaces play a crucial role. A C++ code is provided which also augments the karambola package for the computation of Minkowski tensors on triangulated surfaces. In particular, the elaborated boundary condition for adjacent triangles to guarantee convergence are very practical, especially equation (14) and its implementation in karambola is useful for further applications.
The paper is clearly written and very nicely illustrated by helpful figures.

The possibility to coarse-grain in a well-defined manner is a very important consequence of integral geometry and the advantages of the Minkowski functionals are well emphasised. I particularly liked the motivation and the outlook where this coarse-graining could be applied in biophysics, where the transfer of integral geometry to systems such as membranes and cell tissues is certainly a promising step in the future.

However, the relation to Minkowski tensors is, as far as I understand the appendix, even closer than already stated in the paper. In integral geometry the curvature tensor is defined as the integral of the local support measure over a finite domain of the normal bundle, thus it is local and even on edges well-defined. The selection of a region or patch - as it is done here in the paper - is practical for applications, but as far as I can see not a fundamental new extension of the existing literature. In integral geometry curvature measures are introduced locally and are identical to the speciual case of Minkowski tensors only when integrated over a closed surface.

Unfortunately, I have not been able to understand the remark that the introduced curvature tensor is not continuous. Even if normal vectors at contours are not uniquely given, one can usually apply a Gauss map (on the parallel surface) and define local Minkowski tensors by integrals over a part of the possible normal vectors. Therefore, I would appreciate a more detailed proof since I do not see at the moment why it should not be a local Minkowski tensor; although certainly a different one than the curvature weighted integral over the patch. Similar curvatures tensors at singular edges are used, for instance, in density functional theories for fluids with hard particle interactions; see e.g. Hansen-Goos&Mecke, PRL 102, 018302 (2009); J. Phys.: Condens. Matter 22, 364107 (2010).

There are minor issues which might be changed to improve the paper:
The authors should add a classic reference to integral geometry, like the book by Schneider and Weil, 'Stochastic and Integral Geometry' (Springer, 2008) or the paper of Daniel Hug, 'Measures, curvatures and currents in convex geometry' (Freiburg, 1999).

Remarks on page 2:
'Despite its importance, a proper definition of the curvature tensor remains elusive for surfaces represented by discrete triangles or polygons.'
This is not really correct; since the early days of computers there is vast number of publications in computational geometry on proper definitions of discretized curvatures. In particular, convex geometry has solved the problem of curvature tensors on polygons already in a rigorous way, for instance through the definition of (flag) support measures and local valuations for convex bodies including polytopes (see the work by Daniel Hug).

'Minkowski functionals are defined for closed surfaces.'
This seems to be a too restrictive since local functionals can also be defined for open sets (see the reviews mentioned above).

'... and form a basis for scalar and tensor-valued valuations on convex shapes'
While the scalar Minkowski functionals form a basis for scalar valuations, the statement is not quite correct for tensor-valued valuations since some of the Minkowski tensors are linearly depended. Therefore "span the space of" is more precise than "form a basis for".
Moreover, I suggest to add here citations of the theorems proven by Hadwiger (scalar case) and Alesker (tensorial case):
S. Alesker, Geom. Dedicata 74 , 241 (1999); S. Alesker, 'Continuous rotation invariant valuations on convex sets', Annals of Mathematics, Bd. 149, 1999, S. 977-1005.
H. Hadwiger, Abhdl. Math. Sem. Hamburg 17, 69 (1951).

'... are a generalization of the Minkowski scalars to tensorial quantities'.
I suggest to mention here also the application of Minkowski tensors on cellular and discretized structures in physics, for instance, the review article by Schröder-Turk et al., Advanced Materials 23, 2535-2553 (2011).
In particular, the definition of Minkowski maps for local curvature tensors (see Fig. 4 in this reference) might be important for further applications, since it provides a coarse grained evaluation of a Minkowski tensors for pixelated images and polygonal surfaces defined on grids.

p4: '... is a Minkowski tensor'
It might help to add which tensor is addressed, i.e., W_2^{0,2}.

p5: Typo: 'closed surface S' --> 'closed surface \mathcal{S}'

p7: 'Indeed, imposing that ...'
Unfortunately I don't understand the need for this restriction. The authors could explain in more detail, why is it important for the integrated tensor that two neighbouring triangles are assigned the same value?.

p13/14: Typo: 'convention for the integrated curvature tensor that read:'

  • validity: high
  • significance: good
  • originality: good
  • clarity: high
  • formatting: excellent
  • grammar: excellent

Author:  Charlie Duclut  on 2021-11-03  [id 1906]

(in reply to Report 1 on 2021-06-08)

We thank the Referee for their careful reading of our manuscript and their constructive comments. The points raised by the Referee helped us to significantly improve our manuscript and to correct some misleading statements. We provide below a detailed response to all points raised by the Referee.

Answers to the Referee

However, the relation to Minkowski tensors is, as far as I understand the appendix, even closer than already stated in the paper. In integral geometry the curvature tensor is defined as the integral of the local support measure over a finite domain of the normal bundle, thus it is local and even on edges well-defined. The selection of a region or patch - as it is done here in the paper - is practical for applications, but as far as I can see not a fundamental new extension of the existing literature. In integral geometry curvature measures are introduced locally and are identical to the speciual case of Minkowski tensors only when integrated over a closed surface.

We thank the Referee for this remark, and we agree that our terminology was not quite appropriate. We have therefore rephrased the relevant parts of the manuscript. In the revised version, we now explain more clearly the relevance of local Minkowski tensors to our work.

Unfortunately, I have not been able to understand the remark that the introduced curvature tensor is not continuous. Even if normal vectors at contours are not uniquely given, one can usually apply a Gauss map (on the parallel surface) and define local Minkowski tensors by integrals over a part of the possible normal vectors. Therefore, I would appreciate a more detailed proof since I do not see at the moment why it should not be a local Minkowski tensor; although certainly a different one than the curvature weighted integral over the patch. Similar curvatures tensors at singular edges are used, for instance, in density functional theories for fluids with hard particle interactions; see e.g. Hansen-Goos&Mecke, PRL 102, 018302 (2009); J. Phys.: Condens. Matter 22, 364107 (2010).

We agree that the discussion regarding the lack of continuity was misleading. We have rewritten the relevant parts and now rather emphasize the difference between the surface and the boundary contributions. We also explain more clearly the unique choice of the weight factor $a_{\mathbf{e}}$.

There are minor issues which might be changed to improve the paper: The authors should add a classic reference to integral geometry, like the book by Schneider and Weil, 'Stochastic and Integral Geometry' (Springer, 2008) or the paper of Daniel Hug, 'Measures, curvatures and currents in convex geometry' (Freiburg, 1999).

We thank the Referee for pointing us to this relevant literature, that we are now citing.

Remarks on page 2: 'Despite its importance, a proper definition of the curvature tensor remains elusive for surfaces represented by discrete triangles or polygons.' This is not really correct; since the early days of computers there is vast number of publications in computational geometry on proper definitions of discretized curvatures. In particular, convex geometry has solved the problem of curvature tensors on polygons already in a rigorous way, for instance through the definition of (flag) support measures and local valuations for convex bodies including polytopes (see the work by Daniel Hug).

We thank the referee for this comment. We now start the paragraph by first highlighting the early work mentioned by the referee on discretized surfaces. We then point to the need for an improvement on how to define robust local curvature tensors on discrete surfaces.

'Minkowski functionals are defined for closed surfaces.' This seems to be a too restrictive since local functionals can also be defined for open sets (see the reviews mentioned above).

We have modified this sentence to emphasize the fact that Minkowski tensors can be defined locally.

'... and form a basis for scalar and tensor-valued valuations on convex shapes' While the scalar Minkowski functionals form a basis for scalar valuations, the statement is not quite correct for tensor-valued valuations since some of the Minkowski tensors are linearly depended. Therefore "span the space of" is more precise than "form a basis for". Moreover, I suggest to add here citations of the theorems proven by Hadwiger (scalar case) and Alesker (tensorial case): S. Alesker, Geom. Dedicata 74 , 241 (1999); S. Alesker, 'Continuous rotation invariant valuations on convex sets', Annals of Mathematics, Bd. 149, 1999, S. 977-1005. H. Hadwiger, Abhdl. Math. Sem. Hamburg 17, 69 (1951).

We thank the referee for pointing this imprecision. We have modified the sentence and added the suggested references.

'... are a generalization of the Minkowski scalars to tensorial quantities'. I suggest to mention here also the application of Minkowski tensors on cellular and discretized structures in physics, for instance, the review article by Schröder-Turk et al., Advanced Materials 23, 2535-2553 (2011). In particular, the definition of Minkowski maps for local curvature tensors (see Fig. 4 in this reference) might be important for further applications, since it provides a coarse grained evaluation of a Minkowski tensors for pixelated images and polygonal surfaces defined on grids.

We have added the reference and the sentence in the revised manuscript.

p4: '... is a Minkowski tensor' It might help to add which tensor is addressed, i.e., $W_2^{0,2}$.

We now explicitly name this tensor.

p5: Typo: 'closed surface $S$' --> 'closed surface $\mathcal{S}$'

We have fixed this typo.

p7: 'Indeed, imposing that ...' Unfortunately I don't understand the need for this restriction. The authors could explain in more detail, why is it important for the integrated tensor that two neighbouring triangles are assigned the same value?.

We agree with the referee and there is in fact no need for a restriction in this case. We have therefore removed this paragraph.

p13/14: Typo: 'convention for the integrated curvature tensor that read:'

We have fixed this typo.

---

## Round 2 · List of Changes

- we have modified our statements regarding continuity throughout the manuscript (in the introduction, in the conclusion and in section I);
- we have changed the discussion regarding the weighting fraction throughout the manuscript (in the introduction, in the conclusion and in section I);
- we have included Eqs. (10) and (11) to the main text;
- we have added references to relevant and seminal work in integral geometry.

You are currently on this page

Resubmission 2104.07988v2 on 9 November 2021

---

## Editorial Decision

published